# BRIDGING SEQUENCE AND STRUCTURE: LATENT DIF­FUSION FOR CONDITIONAL PROTEIN GENERATION

## ABSTRACT

Protein design encompasses a range of challenging tasks, including protein fold­ing, inverse folding, and protein-protein docking. Despite significant progress in this domain, many existing methods address these tasks separately, failing to adequately leverage the joint relationship between protein sequence and three-dimensional structure. In this work, we propose a novel generative modeling technique to capture this joint distribution. Our approach is based on a diffusion model applied on a geometrically-structured latent space, obtained through an en­coder that produces roto-translational invariant representations of the input protein complex. It can be used for any of the aforementioned tasks by using the diffusion model to sample the conditional distribution of interest. Our experiments show that our method outperforms competitors in protein docking and is competitive with state-of-the-art for protein inverse folding. Exhibiting a single model that ex­cels on on both sequence-based and structure-based tasks represents a significant advancement in the field and paves the way for additional applications.

## 1 INTRODUCTION

Machine learning-based innovations in the fields of structural and computational biology, especially the ability to predict protein structure from amino acid sequence ("protein folding") (Wang et al., 2017; Jumper et al., 2021) and vice-versa ("inverse folding") (Dauparas et al., 2022), have led to groundbreaking advancements in understanding protein structure and function. However, while the study of individual proteins provides foundational insights, studying *protein complexes* (systems with multiple interacting proteins) is imperative to discerning the intricacies of cellular processes and disease mechanisms. For example, in drug development, one typically tries to design a new protein that binds to a given target forming a complex (Modell et al., 2016). Protein generation involves a range of challenging tasks, such as folding, inverse folding, and docking (Kuhlman & Bradley, 2019). Traditionally, these tasks are considered as stand-alone problems, each addressed by specially tailored techniques that leverage structure or sequence information (Ferruz et al., 2022).

For instance, protein-protein docking has recently been addressed using diffusions over manifolds (Ketata et al., 2023) and regression-based models (Evans et al., 2021; McPartlon & Xu, 2023), while inverse folding has mostly approached using autoregressive models (Dauparas et al., 2022; Hsu et al., 2022). Moreover, diffusion models designed for protein generation almost always treat the design of structure and sequence as independent tasks, typically by applying an inverse folding model to generated structures (Watson et al., 2023; Trippe et al., 2023). Unfortunately, none of these methods holistically address protein generation. By addressing these tasks individually, these methods do not adequately leverage the joint relationship of protein sequence and three-dimensional structure.

In this work, we introduce OMNIPROT (fig. 1), a generative model that inherently captures this re­lationship and can tackle the diverse set of tasks arising in protein generation in a unified way. OM­NIPROT has two main components: an autoencoder with a geometrically-structured latent space, and a diffusion model that operates in this latent space. OMNIPROT's autoencoder is tailored to protein complexes, leveraging roto-translational invariant features to produce roto-translational in­variant latent representations that jointly capture sequence and structural information. By designing the diffusion to operate in this latent space, OMNIPROT can be seamlessly used to address *any* con­ditional generative task in protein design in a unified way, from (full-atom) flexible protein-protein

docking to inverse folding, by simply changing the conditioning information provided to the latent diffusion model.

We evaluate OMNIPROT on two common tasks from conditional generative protein design, protein-protein docking and inverse folding. Notably, in protein-protein docking, OMNIPROT attains a 70% DockQ success rate, compared to 63.3% of the second best ML-based approach. In addition, OMNIPROT achieves a sequence recovery rate of 46.8% on the inverse folding PROTEINMPNN benchmark (Dauparas et al., 2022), which is close to the 48.8% achieved by PROTEINMPNN, the state of the art method tailored for this task. We believe OMNIPROT represents a significant step towards a unified approach for protein generation and design.

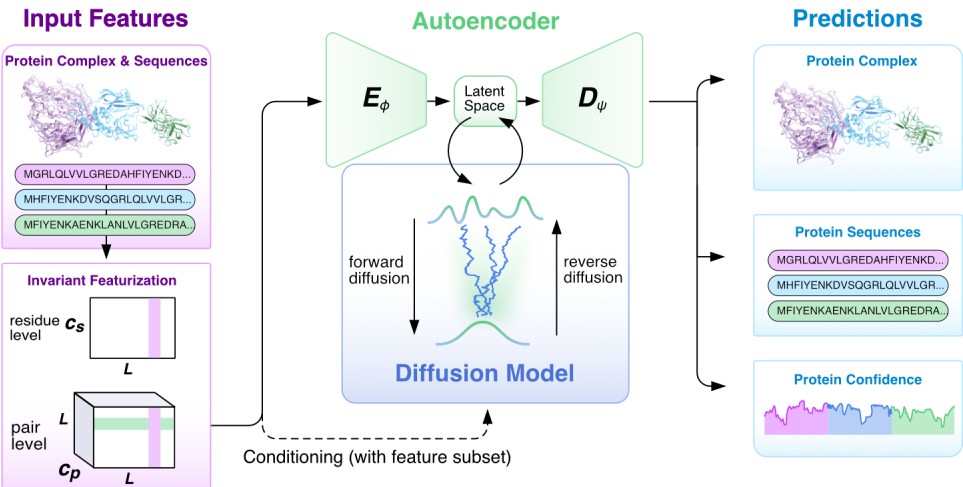

Figure 1: **OMNIPROT overview**. As explained in section 4.4 OMNIPROT provides a unified way of addressing conditional protein generation tasks (e.g. protein-protein docking, inverse folding, among others), by simply changing the conditioning features provided to the diffusion model.

## 2 PRELIMINARIES - DIFFUSION MODELS

Diffusion models (Sohl-Dickstein et al., 2015; Song & Ermon, 2019; Ho et al., 2020) represent a powerful generative modeling technique. Given a target distribution $p_{\text{data}}(z)$, they define a forward process that gradually transforms this distribution into a tractable reference. For instance, the variance preserving formulation from Song et al. (2020) defines this process using an SDE,

$$\mathrm{d}z_t = -\tfrac{1}{2}\beta(t)\,z_t\mathrm{d}t + \sqrt{\beta(t)}\mathrm{d}w, \quad \text{where} \quad t \in [0,1] \quad \text{and} \quad z_0 \sim p_{\text{data}}(z). \tag{1}$$

Essentially, this process takes a dataset of samples from $p_{\text{data}}(z)$ and progressively transforms them into random noise. Critically, it can be simulated exactly for any time $t$: Given $z_0 \sim p_{\text{data}}$ we have

$$z_t \sim p_t(z_t \mid z_0) = \mathcal{N}\left(z_t \ \middle| \ z_0\,e^{-\frac{1}{2}\int_0^t \beta(s)\mathrm{d}s}, I - I\,e^{-\int_0^t \beta(s)\mathrm{d}s}\right). \tag{2}$$

For an appropriate choice for $\beta(t)$ (Song et al., 2020), this shows that samples $z_1$ (obtained by running eq. (1) up to time $t = 1$) approximately satisfy $z_1 \sim \mathcal{N}(0, I)$. Therefore, new samples from $p_{\text{data}}$ can be obtained by simulating the time-reversal (Anderson, 1982) of eq. (1), given by

$$\mathrm{d}z_t = -\frac{\beta(t)}{2}\Big(z_t + 2\,\nabla \log p_t(z_t)\Big)\mathrm{d}t + \sqrt{\beta(t)}\mathrm{d}\bar{w}, \quad z_1 \sim \mathcal{N}(0, I), \tag{3}$$

from $t = 1$ to $t = 0$. Unfortunately, the "score" $\nabla \log p_t(z_t)$ is often intractable. Diffusion models address this training a *score network* $s_\theta(z_t, t)$ to approximate it, minimizing the denoising score matching objective (Hyvärinen & Dayan, 2005; Vincent, 2011)

$$\mathcal{L}(\theta) = \mathbb{E}_{t,z_0,z_t \mid z_0}\left[w(t)\,\|s_\theta(z_t, t) - \nabla_{z_t} \log p_t(z_t \mid z_0)\|^2\right]. \tag{4}$$

Finally, new samples from $p_{\text{data}}(z)$ can be obtained (approximately) by simulating the reverse process from eq. (3) using $s_{\theta*}(z_t, t) \approx \nabla \log p_t(z_t)$.

**Conditional diffusion models** are a natural extension of the formulation above, in which a diffusion model is trained to approximate conditional distributions $p_{\text{data}}(z\,|c)$, where $c$ is the conditioning variable. In this case, the dataset consists on tuples $(z, c)$, the score network is given by $s_\theta(z_t, t, c)$, and the reverse process produces samples from $p_{\text{data}}(z\,|\,c)$ (for any given $c$).

## 3   RELATED WORK – MACHINE LEARNING FOR PROTEIN DESIGN

**Conditional protein design** represents a broad domain that encompasses multiple tasks, each addressing distinct yet interconnected aspects of protein structure and sequence generation. This section briefly reviews machine learning based approaches for many of these tasks.

**Protein folding** attempts to predict the 3D conformation of a a protein given its sequence of amino acids (residues). Deep learning-based methods have recently achieved impressive performance on this task (Jumper et al., 2021; Lin et al., 2023; Mirdita et al., 2022; Yang et al., 2020). However, they were originally designed to produce single conformations, failing to model protein flexibility (Lane, 2023). Recent works tried to bridge this gap (Stein & Mchaourab, 2022), with a promising direction involving the use of diffusion models to sample multiple conformations (Jing et al., 2023).

**Inverse folding** aims to predict a protein sequence from its 3D structure. Recent deep learning methods provide efficient alternatives to physics-based approaches, spanning autoregressive models (Ingraham et al., 2019; McPartlon et al., 2022; Anand et al., 2022) and other specialized architectures (Qi & Zhang, 2020; Zhang et al., 2020; Jing et al., 2020; Strokach et al., 2020; Gao et al., 2023; Hsu et al., 2022). Sequence and structure co-design methods are similarly promising (Shi et al., 2023).

Machine learning has also reached the realm of **protein-protein docking**, with multiple approaches introduced to lighten the computational costs of traditional physics-based methods (Chen et al., 2003; De Vries et al., 2010; Yan et al., 2020). Recently proposed deep learning methods that build on transformer models (McPartlon & Xu, 2023), diffusion models Ketata et al. (2023), and attention-based networks (Evans et al., 2021) have achieved impressive performances on this task.

In recent years, **diffusion models** have been increasingly used for protein generation. Several variants have been proposed for backbone generation, diffusing over coordinates (Trippe et al., 2023), inter-residue angles (Wu et al., 2022), or residue frames position and orientation (Lin & AlQuraishi, 2023; Yim et al., 2023). Some methods diffuse over sequence and structure jointly, producing sequence and backbone (Lisanza et al., 2023), and side-chain (Anand & Achim, 2022) atoms.

Finally, **latent diffusion models** (LDMs) have been used for protein sequence synthesis (Jiang et al., 2023), protein backbone (Fu et al., 2023) and 3D molecule generation (Xu et al., 2023). All methods employ latent diffusion, using different approaches to construct the continuous latent space to encode input information (e.g. the latter splits the latent space into invariant and equivariant components to encode molecules). These works displayed good performances on their respective tasks, demonstrating potential benefits of using latent diffusion for generative tasks in structural biology.

## 4   LATENT DIFFUSION FOR JOINT SEQUENCE-STRUCTURE LEARNING

This section introduces OMNIPROT (fig. 1), a generative model able to jointly capture protein sequence and three-dimensional structure. It consists of a latent diffusion model (Rombach et al., 2022; Vahdat et al., 2021) operating on the geometrically structured latent space of a pre-trained protein autoencoder. The training of OMNIPROT follows a two-stage approach (Rombach et al., 2022). First, we train an autoencoder tailored for protein complexes (described in section 4.2), using roto-translational invariant features (section 4.1). The autoencoder's latent space effectively captures both the sequence and structural information of the input protein complex. Second, with the autoencoder frozen, we train a diffusion model that operates in this latent space (section 4.3). To showcase the OMNIPROT's versatility, section 4.4 delineates its application across various generative protein design tasks, from inverse folding to protein-protein docking.

### 4.1   ROTO-TRANSLATIONAL INVARIANT FEATURES FOR PROTEIN COMPLEXES

Given a protein complex, we extract roto-translational invariant features to coarsely characterize its sequence and structure. These features are then used to train the autoencoder and the latent diffusion.

We follow the features used by DockGPT (McPartlon & Xu, 2023), which are split into three types: **residue-level** (information for each residue $i$ in isolation of other residues, including amino acid type, sequence position, and backbone angles), **intra-chain-pair** (information for pairs of residues $i, j$ in the same chain, including their distance, relative orientation, and sequence separation), and **inter-chain-pair** (information for pairs of residues $i, j$ in different chains, including their distance, relative orientation, contact information, and relative chain information).

For each residue $i$, or residue pairs $i, j$, all the aforementioned features are one-dimensional arrays. For instance, the residue-level amino acid type is a one-hot vector over the 20 natural amino acids, while the inter-chain-pair distance is a one-hot vector obtained by binning distances between residues into bins of width 2Å. We provide details on how all these features are generated in appendix B. Given a protein complex with $L$ residues, residue-level features are combined into an $L \times c_s$ matrix $\mathbf{s}$, and intra- and inter-chain-pair features are combined into an $L \times L \times c_p$ tensor $\mathbf{p}$.

## 4.2 AUTOENCODER WITH ROTO-TRANSLATIONAL INVARIANT LATENT SPACE

The first component of OMNIPROT is an **autoencoder**, consisting of a stochastic **encoder** (which maps a protein complex to a roto-translational invariant latent representation that jointly captures sequence and structure) and a **decoder** (which reconstructs the input complex, both sequence and structure, given its latent representation). As detailed in appendix A, the autoencoder is trained independently of the diffusion, by minimizing a combination of the reconstruction loss (for the predicted structure) and the cross-entropy loss (for the predicted sequence).

**Encoder $\mathcal{E}_\phi$** Given a protein complex, the encoder computes the mean and variance of a Gaussian distribution over the latent space, which is sampled to produce the latent representation. Each layer in the encoder is given by (we use 8 layers)

$$\mathbf{s}_i \leftarrow \text{PairBiasAtt}(\mathbf{s}, \mathbf{p}_{i:}), \tag{5}$$

where $\mathbf{s}$ and $\mathbf{p}$ are the residue-level and pair features, respectively, and $\text{PairBiasAtt}$ is the pair-biased attention layer from Jumper et al. (2021). [1] The mean and log-scale of the Gaussian distribution are then obtained as

$$\mu_i \leftarrow \text{Linear}(\mathbf{s}_i), \quad \log \sigma_i \leftarrow \text{Linear}(\mathbf{s}_i). \tag{6}$$

Finally, the latent representation $\mathbf{z}$ (an $L \times 16$ matrix) is obtained as $\mathbf{z}_i = (\tilde{\mathbf{z}}_i - \text{mean}(\tilde{\mathbf{z}}_i))/\text{std}(\tilde{\mathbf{z}}_i)$, where $\tilde{\mathbf{z}} \sim \mathcal{N}(\mu, \sigma^2)$. This latent representation jointly captures structural and sequence information for each residue in the input complex.

**Decoder $\mathcal{D}_\psi$** Given a latent representation $\mathbf{z}$, the decoder is designed to reconstruct the input protein complex (sequence and structure) together with a confidence score for its prediction. We use the structure module from AlphaFold2 (Jumper et al., 2021), where each residue in the reconstructed backbone (represented as the frame formed by the $N$-$C_\alpha$-$C$ atoms) is identified with a rigid transformation $\mathbf{T}_i$ consisting of a translation and a rotation. Using $\mathbf{q}$ to denote the $L \times L \times c_q$ tensor obtained by combining two *pair* features (sequence separation and relative chain information, see section 4.1), each layer in the decoder is given by (we use 8 layers)

$$\mathbf{z}_i \leftarrow \text{IPA}(\mathbf{z}, \mathbf{T}_{:}, \mathbf{q}_{i:}), \quad \mathbf{z}_i \leftarrow \text{MLP}(\mathbf{z}_i), \quad \mathbf{T}_i \leftarrow \mathbf{T}_i \circ \text{BackboneUpdate}(\mathbf{z}_i). \tag{7}$$

The invariant point attention (IPA) and backbone update operations are described in detail in Jumper et al. (2021). The side chain angles, amino acid type (logits over the 20 natural amino acids), and confidence score are then predicted as (for each residue $i$ in the backbone)

$$\text{angles}_i = \text{MLP}(\mathbf{z}_i), \quad \text{aa}_i = \text{Linear}(\mathbf{z}_i), \quad \text{conf}_i = \text{Linear}(\mathbf{z}_i). \tag{8}$$

## 4.3 LATENT SPACE DIFFUSION

The second component of OMNIPROT is a **conditional diffusion model** operating in the autoencoder latent space. This diffusion is trained in a second step, after training and freezing the autoencoder (Rombach et al., 2022).

---

[1]This is a self-attention mechanism (Vaswani et al., 2017) on $\mathbf{s}$, with an extra bias term for the dot-product affinity between each pair of residues $i, j$, computed using $\mathbf{p}_{ij}$.

As detailed in section 2, diffusion models define a forward process that gradually diffuses samples $\mathbf{z}$ (in our case latent representations) by running a forward diffusion process ("noising"). Then, they generate samples by reversing this process ("denoising"). This requires training a *score network* $s_\theta(z_t, t, c)$, where $c$ represents conditioning information available to the model. In our case, this may include any subset of the *residue-level* and *pair-level* features described in section 4.1. Critically, as detailed in section 4.4 (and summarized in table 1), the exact features included in $c$ will depend on the task being addressed. For instance, for protein-protein docking, $c$ will include all *residue-level* and *intra-chain-pair* features, but no *inter-chain-pair* features (i.e. no information of interactions between different chains); for inverse folding, on the other hand, $c$ will only contain features related to protein structure, without including information regarding amino acid types.

**Score network** Our score network $s_\theta(\mathbf{z}_t, t, c)$ resembles the encoder architecture, with extra updates for the *pair* features through triangular multiplicative layers (Jumper et al., 2021). Using $\mathbf{u}^t$ to denote the $L \times (16 + c_u)$ matrix obtained by concatenating $\mathbf{z}_t$ and the *residue-level* features in $c$, and $\mathbf{r}$ to denote the $L \times L \times c_r$ tensor obtained by combining *pair-level* features in $c$, each layer in the score network consists of (we use 12 layers)

$$\mathbf{u}_i^t \leftarrow \text{PairBiasAtt}^*(\mathbf{u}^t, \mathbf{r}_{i:}, t_{\text{enc}}), \quad \mathbf{r}_{ij} \leftarrow \mathbf{r}_{ij} + \text{OutSum}(\mathbf{u}_i^t, \mathbf{u}_j^t), \quad \mathbf{r} \leftarrow \mathbf{r} + \text{TriangMult}(\mathbf{r}), \tag{9}$$

where $t_{\text{enc}}$ denotes the sinusoidal encoding of $t$ (Vaswani et al., 2017), and $\text{PairBiasAtt}^*$ is a variant of the pair-biased attention layer (Jumper et al., 2021) that uses $t_{\text{enc}}$ to compute attention weights (appendix D). The final score is obtained through a linear layer $\text{score}_i = \text{Linear}(\mathbf{u}_i)$.

### 4.4 A UNIFIED APPROACH FOR CONDITIONAL PROTEIN GENERATION TASKS

Conditional protein generation using OMNIPROT is done by running the diffusion model (conditioned on a subset of structural and sequence features $c$) in the latent space, and feeding the resulting sample through the decoder, which produces both sequence and structure. OMNIPROT can be flexibly used for multiple conditional generative tasks, by simply selecting an appropriate subset of features (section 4.1) to use at inference time. Table 1 presents a comprehensive summary of various tasks along with their corresponding feature requirements. Concurrently, fig. E.1 illustrates a specific example of the OMNIPROT pipeline for protein-protein docking.

| Task | Features used as conditioning information c by task | | | |
| | Protein Sequence | Intra-Chain Geometry | Inter-Chain Geometry | Residue Contacts |
|---|---|---|---|---|
| Docking with contacts | ✓ | ✓ | | ✓ |
| Docking without contacts | ✓ | ✓ | | |
| Folding | ✓ | | | |
| Inverse Folding | | ✓ | ✓ | ✓ |

Table 1: **Summary of conditional generation tasks** addressed by OMNIPROT with the subset of features (section 4.1) used by each one as conditioning information.

**Inverse folding.** If $c$ includes features that contain three-dimensional structural information, but no information about amino acid types, OMNIPROT will sample sequences compatible with the provided three-dimensional structure, effectively performing inverse folding.

**Blind protein-protein docking.** If $c$ includes *residue-level* and *intra-chain-pair* features, without any *inter-chain-pair* features (i.e. no information of interactions between different chains), OMNIPROT will sample three-dimensional structures for the full protein complex, effectively performing flexible (blind) protein-protein docking.

**Protein-protein docking with contact information.** Additionally, by including some inter-chain contact points in $c$ (contact information *inter-chain-pair* features), OMNIPROT will leverage this information to sample three-dimensional structures for the full protein complex, effectively performing flexible protein-protein docking with contact information. Having such contact information available from experimental or design constraints is a common scenario, known as integrative modeling, and has long standing in the field (Koukos & Bonvin, 2020).

**Protein folding.** When working with a single protein, if $c$ does not include any structural-related features, OMNIPROT will sample three-dimensional structures compatible with the provided sequence, effectively performing flexible protein folding.

### 4.4.1 OMNIPROT ANALYSIS

After training, OMNIPROT's sampling process (latent diffusion + decoder) defines a distribution over protein complexes, structure and sequence. This section briefly studies this distribution's properties.

**Proposition 1.** *Let $(x, s)$ denote a protein structure and sequence, and $p(x, s \mid c)$ denote the distribution defined by* OMNIPROT*, where $c$ is the conditioning information provided to the diffusion model. If the decoder initializes backbone frames with a global rotation chosen uniformly at random, then $p(x, s \mid c) = p(R\, x, s \mid c)$ for any three-dimensional rotation $R$.*

We prove proposition 1 in appendix C. The proposition states that the joint distribution over sequence and structure defined by OMNIPROT, denoted by $p(x, s \mid c)$, is invariant w.r.t. rotations of the structure. Furthermore, this holds regardless of the subset of features used as the conditioning information $c$. This is a desirable property, as the true data distribution satisfies this invariance. (It has been observed that methods that enforce invariances and equivariances present in the true data distribution sometimes yield better generalization (Jumper et al., 2021; Xu et al., 2022).)

Another desirable property satisfied by OMNIPROT's distribution is its invariance w.r.t. rigid transformations of the structural information included as conditioning information. The relevance of this can be seen through concrete application examples. For inverse folding, OMNIPROT's predicted distribution over sequences should not be affected by rigid transformations of the structural information provided as input. For (blind) protein-protein docking, the distribution over full complex structure should not be affected by rigid transformations of the individual chains' structures provided as input. OMNIPROT satisfies these invariances. This follows directly from the fact that it only relies on roto-translational invariant features (section 4.1), which are unaffected by rigid-body transformations.

## 5 EMPIRICAL EVALUATION

Although OMNIPROT is capable of performing any protein generation task, we evaluate it on inverse folding and protein docking, as these are at the core of conditional protein generation. We first introduce the datasets and metrics used for each task, and then present our empirical results. In all tables, we use use bold to denote the best performing method, and underline the second-best.

**Protein-protein docking.** Our dataset contains all available chains in the Protein Data Bank (PDB, March 2023, 199k proteins). Splits are generated by performing FoldSeek all-vs-all structural alignments of protein binding sites (Berman et al., 2003; van Kempen et al., 2023). This novel split is introduced to address significant potential data leakage found in the DIPS splits used in previous rigid docking methods (Ganea et al., 2021; Ketata et al., 2023), where large fractions of the test data contained structural overlap with the training data. The full detail with further evidence of the necessity of these new splits are provided in appendix F. Possible test set candidates are selected from cluster representatives of the clusters with the top 10% highest resolution, which contained at least one high quality representative protein-protein interaction. Out of these candidates, we randomly chose a subset of 150 dimers, 100 heterodimers and 50 homodimers. The training data (199k proteins) consists of the remaining data without any quality-based filtering.

We evaluate protein-protein docking methods by measuring differences between predicted and ground truth structures in terms of root mean square deviation (RMSD), RMSD for interface residues (I-RMSD), and RMSD for ligand residues (L-RMSD). For RMSD, we report 25th and 50th percentiles, and the proportion of predictions with I-RMSD $\leq$3Å and L-RMSD$\leq$6Å. Further, we report DockQ, which is a composite score of I-RMSD, C-RMSD and Fnat (fraction of recovered native contacts) (Basu & Wallner, 2016). The continuous DockQ score (range 0 to 1) can be used to reproduce the Critical Assessment of PRediction of Interactions (CAPRI) classification of Incorrect, Acceptable, Medium and High quality predictions (Vajda et al., 2002).

**Inverse folding.** We use the dataset curated for the development of PROTEINMPNN (Dauparas et al., 2022). It contains protein assemblies in the PDB (Berman et al., 2003) (as of Aug 02, 2021) clustered by 30% sequence identity using mmseqs2 (Steinegger & Söding, 2017). We evaluate methods on a test set of 150 monomers selected at random (from the original test set). Additional information regarding the dataset and data collection can be found in appendices F.2 and G.

We measure inverse folding performance using using native sequence recovery rate (NSR), and self-consistency RMSD (sc-RMSD) (Trippe et al., 2023) with ESMFold structures (Lin et al., 2023). The rationale is that NSR assesses how closely a designed sequence matches the native sequence of an input backbone structure, and sc-RMSD provides an in-silico estimate of how well sequences encode structure by measuring the RMSD between predicted and ground-truth backbones. In accordance with Dauparas et al. (2022); Jing et al. (2021); McPartlon et al. (2022), NSR is reported as the median (over all structures) of the average percentage of residues recovered correctly. Further, we report the proportion of predictions with sc-RMSD$\leq$ 2.5Å and $\leq$5Å.

## 5.1 AUTOENCODER EVALUATION

We begin our empirical evaluation by studying the autoencoder's accuracy, as it may limit OMNIPROT's performance on downstream tasks. We do this by simply measuring its capacity to reconstruct input complexes, both structure and sequence, using the metrics described above. We found that OMNIPROT's autoencoder achieves an NSR value of $98\pm1\%$ for sequence recovery. For structure recovery, samples from OMNIPROT's encoder are decoded with an average full-atom Complex RMSD of $1.3 \pm 0.3$Å, average I-RMSD is $1.3 \pm 0.3$Å, and L-RMSD is $2.1 \pm 0.5$Å. With these RMSD statistics, the average DockQ score of predicted complexes is $0.75 \pm 0.05$Å, which is at the upper threshold of medium quality. Although recovered structures are of relatively high quality, in section 5.2, we show that OMNIPROT is capable of generating structures at the lower bound of the autoencoder's recovery range. This suggests that improvements to the autoencoder could directly translate to performance gains on design tasks.

## 5.2 PROTEIN-PROTEIN DOCKING

We compare OMNIPROT to three machine learning approaches for protein-protein docking, the regression-based methods EQUIDOCK (Ganea et al., 2021) and DOCKGPT (McPartlon & Xu, 2023), and the diffusion-based method DIFFDOCK-PP (Ketata et al., 2023). We re-trained each of these methods on the dataset described above. Training details for baselines are provided in appendix G.

| | I-RMSD(Å)↓ | | | L-RMSD(Å)↓ | | | DockQ↑ | | |
|---|---|---|---|---|---|---|---|---|---|
| | 25th | 50th | %≤ 3Å↑ | 25th | 50th | %≤ 6Å↑ | ≥accep. | ≥med. | ≥high |
| EQUIDOCK | 14.5 | 18.14 | 0.0% | 29.3 | 35.0 | 0.0% | 0.0% | 0.0% | 0.0% |
| DOCKGPT | **0.76** | 2.13 | 55.3% | **1.86** | 5.96 | 50.1% | 63.3% | 52.0% | **31.3%** |
| DIFFDOCK-PP (20)[†] | 2.63 | 5.01 | 31.3% | 6.1 | 13.2 | 24.6% | 44.6% | 24.0% | 4.6% |
| OMNIPROT (20)[†] | 1.45 | **1.92** | **64.7%** | 2.61 | **3.82** | **60.7%** | **70.0%** | **56.7%** | 1.3% |
| OMNIPROT + 1C (20)[†] | 1.31 | 1.50 | 94.0% | 2.11 | 2.73 | 92.0% | 97.3% | 90.7% | 3.3% |

Table 2: **Results on 150 protein dimers**. Results for four ML-based docking methods are shown for the test set. Here, we use 25th and 50th to denote 25th and 50th percentile values. Each method was re-trained and evaluated on the same splits. For diffusion models, the number of sampled poses is shown in parentheses. In an effort to fairly compare our method with DIFFDOCK-PP, we report only oracle statistics, denoted with [†], which refers to the setting where we can perfectly select the best pose out of the sampled ones. We distinguish our performance on blind docking (OMNIPROT) and our performance on site-conditioned docking given one $C_\alpha$-$C_\alpha$ contact (OMNIPROT + 1C)

Table 2 reports the results achieved by all methods on the 150 dimers in the test set. When reporting results for generative methods, OMNIPROT and DIFFDOCK-PP, we sample 20 structures per target and report "oracle" statistics, by selecting the prediction with the lowest RMSD from the ground truth. Although this biases performance in favor of diffusion models, it provides clear and simple criteria that is easy to apply across both methods. (Regression based methods are deterministic, producing a single structure per target.) As an ablation study, we also measure performance for a varying number of sampled structures (5, 10, 20), with results shown in table H.1.

Table 2 shows that OMNIPROT achieves competitive lower-quartile I-RMSD and L-RMSD with DOCKGPT, and significantly outperforms DIFFDOCK-PP and EQUIDOCK on all metrics. In terms of DockQ score, OMNIPROT finds the largest fraction of medium or better quality poses, but falls short of DOCKGPT in terms of high-quality predictions. This is not surprising given that the autoencoder has an average DockQ of 0.75 – marginally below the high quality threshold. We expect improvements to the autoencoder to translate to improvements in OMNIPROT's performance.

We also evaluate OMNIPROT's and DIFFDOCK-PP's performance when generating a different number of samples per target. Results are shown in table H.1. We observe that OMNIPROT tends to converge on low-RMSD solutions with significantly less samples than DIFFDOCK-PP. In fact, OMNIPROT with 5 samples per target significantly outperforms DIFFDOCK-PP with 20 samples *across all metrics*. Considering the best pose across five samples, OMNIPROT achieves median oracle I-RMSD of 2.37Å and median oracle L-RMSD of 5.53Å. Given the same number of samples, DIFFDOCK-PP's median I-RMSD and L-RMSD is 8.67Åand L-RMSD is 19.78Å.

We also assess OMNIPROT's ability to incorporate binding site information in the form of pairwise $C_\alpha$ contacts (included as the contact information inter-chain-pair feature). In line with the results in McPartlon & Xu (2023), we observe that providing even a single inter-chain contact significantly improves docking performance fig. H.1. In fact, with a single contact, OMNIPROT achieves an oracle (out of 20 samples), 90% of OMNIPROT's predictions achieve a medium or high DockQ score. We remark that the 25-th percentile I-RMSD is roughly equal to the error rate of the autoencoder, showing again that OMNIPROT's is performing at the limit imposed by the autoencoder, and that improvements made to the autoencoder could directly translate to improvements in OMNIPROT's performance.

## 5.3 INVERSE FOLDING

We compare OMNIPROT against PROTEINMPNN (Dauparas et al., 2022), the state of the art method for inverse folding. Additionally, to assess the utility of jointly encoding sequence and structure in OMNIPROT's latent representation, we also compare against OMNIPROT(Seq), a variant of OM-NIPROT based on an autoencoder trained *only* on sequence-based features (same architecture).

| | NSR(%) | | sc-RMSD | | |
|---|---|---|---|---|---|
| | Median↑ | Std | Median (Å) ↓ | %≤ 2.5Å↑ | %≤ 5Å↑ |
| PROTEINMPNN | **48.8**% | 0.08 | **1.82** | **62.3** % | **78.8** % |
| OMNIPROT(Seq) | 43.2% | 0.06 | 2.65 | 45.8% | 65.5 % |
| OMNIPROT | 46.8% | 0.06 | 2.34 | 53.6 % | 69.5 % |
| Native | — | — | 2.11 | 59.7% | 74.4% |

Table 3: **Inverse folding results on PDB monomer test targets**. Columns show median and standard deviation of NSR and sc-RMSD statistics for each method (row). For sc-RMSD we show the percentage of predicted structures below a 2.5Å and 5Å cutoff. We add an additional row (Native) showing sc-RMSD statistics for native structures.

Table 3 shows median NSR and sc-RMSD for OMNIPROT, OMNIPROT(Seq), and PROTEINMPNN. Since several authors have reported a correlation between *de novo* design success rate and RMSD (Cao et al., 2022; Watson et al., 2023), we also report the fraction of sequences resulting in predicted structures having at most 2.5Å and 5Å RMSD from native. Overall, we observe that OMNIPROT is competitive with PROTEINMPNN in terms of NSR, with a median of 46.8% recovery, against PRO-TEINMPNN's 48.8%. The performance is similar for sc-RMSD, where OMNIPROT has a median of 2.34Å compared to 1.82Å for PROTEINMPNN. Interestingly, we also observe that OMNIPROT outperforms OMNIPROT(Seq) by a noticeable margin, with OMNIPROT(Seq) sequence recovery dropping 3%, and roughly 15% fewer predicted structures structures falling below a 2.5Å RMSD cutoff. *This shows the benefits of jointly encoding sequence and structure, as done by OMNIPROT, instead of handling either one in isolation.*

A more detailed comparison of sc-RMSD is shown in fig. 2, which contains scatter plots between OMNIPROT and PROTEINMPNN (fig. 2A), and OMNIPROT against OMNIPROT(Seq) (fig. 2B). Reinforcing our previous conclusion regarding the benefits of jointly encoding sequence and structure,

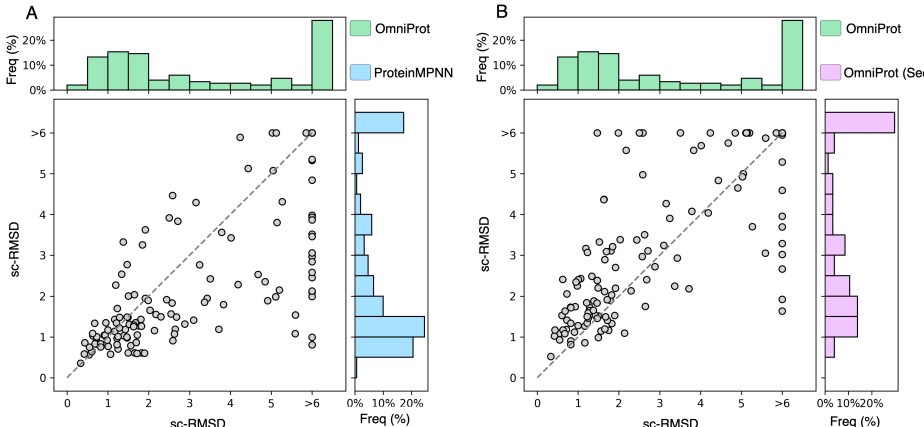

Figure 2: **ESMFold sc-RMSD for OMNIPROT and PROTEINMPNN** designed sequences. Each subfigure shows a scatter plot and histograms of sc-RMSD for 150 monomeric targets from the PDB dataset. (A) comparison of OMNIPROT ($x$-axis) and PROTEINMPNN ($y$-axis). (B) Comparison of OMNIPROT and OMNIPROT(Seq), trained only on sequence encodings ($y$-axis).

fig. 2B shows that sequences predicted by OMNIPROT often lead to structures with lower RMSD than those produced byOMNIPROT(Seq) (better for $\approx 80\%$ of the test set).

Fig. 2A shows that OMNIPROT compares favorably to PROTEINMPNN for roughly 30% of the test set in terms of sc-RMSD. Some of this difference may be explained by the ability of autoregressive models to control sampling temperature. It is possible that related techniques for diffusion models, such as self-attention guidance (Hong et al., 2023) or classifier-free guidance (Ho & Salimans, 2022) could improve OMNIPROT's performance on this task. To test this, we explored the use of low temperature sampling (Ingraham et al., 2022). Results are shown in table H.2, where it can be observed that using a temperature $< 1$ is often beneficial for OMNIPROT. Finally, we remark that NSR and sc-RMSD are both only a proxy for sequence designability. Indeed, PROTEINMPNN achieves sc-RMSD statistics favorable to ground-truth sequences (see table 3). Although native sequences obviously encode native structure, when predicted by ESMFold, only 59.7% predicted structures achieved accuracy below 2.5Å.

Although OMNIPROT slightly under-performs PROTEINMPNN, it offers several advantages. First, the sequence predictions by OMNIPROT were generated using 150 reverse diffusion steps. This is in stark contrast to the randomized autoregressive scheme employed by PROTEINMPNN, which requires independent inference steps for each input residue. In addition, OMNIPROT is capable of jointly designing sequence and structure in a one-shot, although this is beyond our current scope.

## 6 CONCLUSION

We introduced OMNIPROT, a method that jointly models protein structure and sequence, and can address *any* conditional generative protein task in a unified way. For the scope of this work, we evaluate it on protein-protein docking and inverse folding, which lie at the core of protein conditional generation. We compare OMNIPROT against baselines tailored to each of these tasks. We observe that our approach achieves state of the art performance in protein-protein docking, and competitive results on inverse folding. To the best of our knowledge, this is the first method to yield (near) state of the art performance in both tasks simultaneously.

Furthermore, we study the benefits of jointly learning structure and sequence instead of each one in isolation. We explicitly evaluate this for inverse folding, and observe that jointly modeling both structure and sequence leads to noticeable performance improvements.

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

# A  AUTOENCODER TRAINING DETAILS

The loss used to train the autoencoder is given by

$$\mathcal{L}_{\text{ae}}(\phi, \psi) = \text{CrossEntropy}(\hat{S}, S) + \text{FAPE}(\hat{X}, X_{\text{true}})+$$
$$10^{-3} \cdot \text{KL}(\mathcal{N}(\mu, \sigma^2) \,\|\, \mathcal{N}(0, I)) + \text{plDDT}(\hat{X}^{C\alpha}, X_{\text{true}}^{C\alpha}), \quad (10)$$

where $\mathcal{N}(\mu, \sigma^2)$ denotes the distribution in the latent space produced by the encoder, $\hat{X}^a$ is the full-atom three-dimensional structure reconstructed by the decoder and indexed by atom type $a$, and $\hat{S}$ is the reconstructed sequence. The final term $\text{plDDT}$ is taken from (Jumper et al. (2021), Supplemental Algorithm 29) The FAPE loss (Jumper et al., 2021) measures the quality of the produced structure by aligning predicted per-residue predicted and ground truth rigid frames. To account for limited or unknown knowledge of binding interfaces in a protein complex, we mask the contact features $E_{\text{contact}}(i, j)$ when producing the pair representation used as input for the encoder with probability $1/2$. Therefore, half of the samples encountered during training do not contain inter-chain contact information. When these features are not masked, we subsample the number of contacts included as $N_{\text{contact}} \sim \text{Geometric}(1/3)$.

# B  ROTO-TRANSLATIONAL INVARIANT FEATURES

**Residue-level features** ($E_{\text{aa}}$(i), $E_{\text{pos}}(i)$, and $E_{\text{angle}}(\theta_i)$) include amino acid type, sequence position, and backbone angles, respectively. $E_{\text{aa}}(i)$ encodes the type of residue $i$ (as a one-hot encoding of the 20 natural amino acids in the autoencoder, or as the residue ESM embedding (Lin et al., 2023) in the diffusion). $E_{\text{pos}}(i)$ encodes the $i$th residue relative sequence position as a one-hot vector using ten equal-width bins. $E_{\text{angle}}(\theta_i)$ encodes the backbones torsional angles $\theta_i \in \{\phi_i, \psi_i\}$ as a one-hot encoding by splitting $\theta \in [-180°, 180°]$ into 18 equal-width bins.

**Intra-chain pair features** ($E_{\text{dist}}(i, j)$, $E_{\text{angle}}(\theta_{ij})$, and $E_{\text{sep}}(i, j)$) include distance, relative orientation, and sequence separation, respectively. $E_{\text{dist}}(i, j)$ bins the distance between the $i$-th residue $C_\alpha$ atom and the $j$-th residue backbone atom $a \in \{N, C_\alpha, C, C_\beta\}$ into six equal-width groups between 2Å and 16Å. $E_{\text{angle}}(\theta_{ij})$ encodes the angles $\theta_{ij} \in \{\phi_{ij}, \psi_{ij}, \omega_{ij}\}$ of pairwise residue orientations (Yang et al., 2020). $E_{\text{sep}}(i, j)$ produces a one-hot encoding of relative sequence separation between residues $i$ and $j$ into 32 classes (McPartlon et al., 2022). The pairwise features for each chain are stacked to form a block-diagonal input matrix with an additional learned parameter filling the missing off-diagonal entries.

**Inter-chain pair features** ($E_{\text{dist}}(i, j)$, $E_{\text{angle}}(\theta_{ij})$, $E_{\text{contact}}(i, j)$, and $E_{\text{chain}}(i, j)$) include distance, relative orientations, contact information, and relative chain information. $E_{\text{contact}}(i, j)$ is a binary flag indicating whether the distance between the $C_\alpha$ atoms of residues $i$ and $j$ is less than 10Å. $E_{\text{chain}}(i, j)$ is a three-class one-hot encoding indicating whether the index of the chain containing residue $i$ is greater than, equal, or less than the index of the chain containing residue $j$. (The distance and angle features are generated as explained above for the *intra-chain-pair* features.)

# C  PROOF OF PROPOSITION 1

*Proof.* Without loss of generality, we assume that both the input and output structures have mean **0**. This follows from the fact that IPA is translation equivariant, and subtracting the structure's center of mass results in an equivalent update to the output. The proposition is a consequence of the architecture used for OMNIPROT's decoder. The updates from the invariant point attention layer (IPA) are invariant to global rigid transformations of the frames, while the backbone update is equivariant to such transformations. As a result, for a fixed latent representation **z**, initializing all frames with the same random rotation and running the decoder is equivalent to initializing the frames with the identity rotation and applying the random rotation on the decoder's output. Since this rotation is chosen uniformly at random, we have $p(x, s \,|\, \mathbf{z}) = p(R\,x, s \,|\, \mathbf{z})$ for any $R$. This is the key property in the derivation below.

Letting $z$ denote the sample produced by the latent diffusion, and $p_{\text{diff}}(\mathbf{z} \mid c)$ its distribution, we have

$$p(R\,x, s \mid c) = \int p(R\,x, s, \mathbf{z} \mid c)\mathrm{d}\mathbf{z} \tag{11}$$

$$= \int p(R\,x, s \mid \mathbf{z}, c)\, p_{\text{diff}}(\mathbf{z} \mid c)\mathrm{d}\mathbf{z} \tag{12}$$

$$= \int p(R\,x, s \mid \mathbf{z})\, p_{\text{diff}}(\mathbf{z} \mid c)\mathrm{d}\mathbf{z} \tag{13}$$

$$= \int p(x, s \mid \mathbf{z})\, p_{\text{diff}}(\mathbf{z} \mid c)\mathrm{d}\mathbf{z} \tag{14}$$

$$= p(x, s \mid c), \tag{15}$$

where eq. (13) uses the fact that, given $\mathbf{z}$, $(x, s)$ is independent of $c$ (i.e. $c$ is only used to generate $\mathbf{z}$ by running the reverse diffusion; given $\mathbf{z}$, the decoder does not use $c$ in any way.) $\qquad\square$

## D PAIR-BIASED ATTENTION WITH TIME ENCODING

To incorporate continuous time into our diffusion model, we augment the input to self attention layers using the adaptive layernorm strategy introduced in Peebles & Xie (2023). More concretely,

$$\text{PairBiasAtt}^*(s, p, t_{\text{enc}}) = \text{PairBiasAtt}(\text{s} \circ \text{MLP}_\beta(\text{t}_{\text{enc}}) + \text{MLP}_\gamma(\text{t}_{\text{enc}})), \text{p}), \tag{16}$$

where $\circ$ denotes element-wise multiplication and $\text{MLP}_\gamma$, $\text{MLP}_\beta$ map from the dimension of the time encoding to the channel dimension of $s$.

## E OMNIPROT PROTEIN-PROTEIN DOCKING

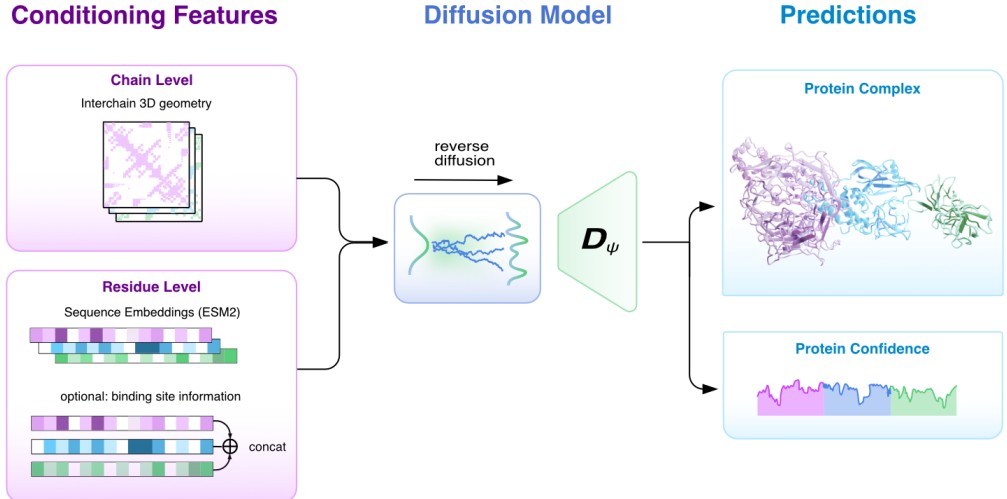

Figure E.1: **High-level overview of protein-protein docking with OMNIPROT.** As explained in section 4.4, OMNIPROT protein-protein docking can be performed blind or with additional per-residue contact information. In both cases conditional features contain ESM2 (3B) protein sequence language model embeddings (Lin et al., 2022), and intra-chain geometry (information for pairs of residues in the same chain, including their distance, relative orientation, and sequence separation). Additional residue features can be included to encode binding site information if available. The versatility of OMNIPROT can easily be leveraged by changing the conditioning features provided to the diffusion model (see table 1).

# F DATASETS

## F.1 PROTEIN-PROTEIN DOCKING

Most recent protein-protein docking methods have been evaluated on the Docking Benchmark 5 (DB5) (Vreven et al., 2015), and trained with complexes from the from Protein Data Bank (PDB) (Berman et al., 2003), such that no protein had more than 30% sequence homology to any protein in the DB5 as proposed by DIPS Townshend et al. (2019). This approach has some limitations. For instance, the size of DB5 is rather small when compared to DIPS, which means the structural diversity of the test set may not be representative, and thus sequence similarity is not always a good proxy to differentiate structurally similar proteins. Even more concerning, however, when performing interface clustering between train, validation and test set of the frequently used DIPS splits Ganea et al. (2021); Ketata et al. (2023), we found that a large majority of the test dataset had structural overlap with the training data set, as evidenced by fig. F.1.

Therefore, to avoid overreporting performance, we train OMNIPROT with splits generated by a structural interface clustering using FoldSeek all-vs-all alignments on all available chains in the PDB (March 2023, 199k proteins), focusing on respective protein binding sites (Berman et al., 2003; van Kempen et al., 2023), and retrain all existing methods on these splits for fair comparison. By using the Foldseek score for clustering, which linearly combines both 3D-based structure and sequence substitution scores (van Kempen et al., 2023), our approach combines sequence and structure-based similarity metrics and restricts them specifically to interfaces. Foldseek stores local alignment positions and normalizes the alignment scores as TM-score, which is used to filter out alignments with lower structural similarity ($< 0.60$ TM-score). Binding site residues were identified based on a criterion of an 6Å $C_\alpha$ distance threshold between chains. A pair of chains was classified as interacting if there were a minimum of 6 binding residues, and at least 50% were encompassed by the Foldseek alignment. Subsequently, a graph representation encoding interface similarity of the interacting chain pairs, where TM-scores served as the weights for the edges, was used to perform community clustering to delineate interface clusters.

The test set consists of the cluster representative with the highest resolution for 10% of the clusters, which contained at least one high quality representative protein-protein interaction (1973 proteins). All representative PPIs in the test set have a minimum resolution of 4.5A, a minimum of 5 atom types, a dimeric state, an interface without any missing residues (gaps), either chain with a maximum of 550 residues and minimum of 25 residues in length, and are solved by X-ray crystallography. The validation set consists of 190 proteins with the same restrictions, except that they may contain gaps. The training data consists of the remaining clusters without any quality-based filtering.

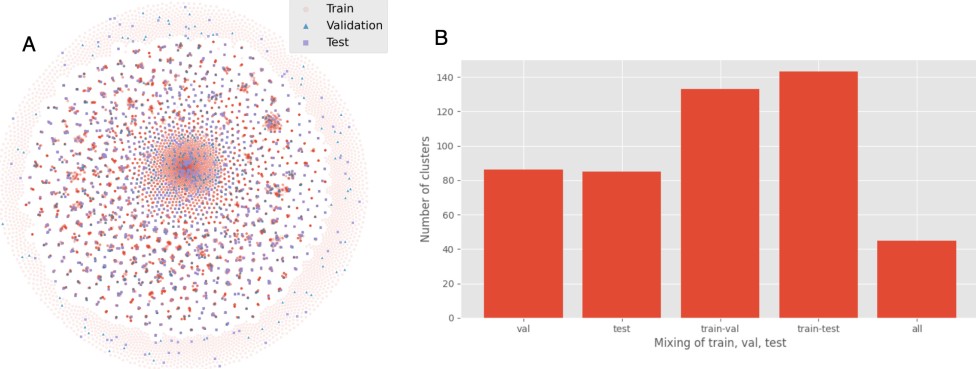

Figure F.1: **Leakage between training, validation and test splits in the DIPS benchmark set (Townshend et al., 2019)**. All-vs-all pairwise structural alignments of respective binding sites performed with Foldseek (van Kempen et al., 2023). (A) TSNE plot of pairwise TM-alignment scores for all chains in DIPS, showing mixed clusters of train (red), validation (blue), and test (purple). (B) Bar plot showing the number of Foldseek clusters against members of DIPS (bars from left to right: only validation, only test, both training and validation, both training and test, or all).

### F.2 INVERSE FOLDING

In generating results for inverse folding we retrained OMNIPROT using the same split as PROTEIN-MPNN. The data set consists 26,361 clusters are split into training (23,358), validation (1,464), and testing (1,539) sets. In accordance with PROTEINMPNN, we randomly sub-sample a single sequence from each cluster at each training epoch. We restricted the test set to 150 randomly selected monomers in order to run sc-RMSD experiments with ESMFold. Results for general proteins were not explored in this work.

## G DATA COLLECTION

To evaluate PROTEINMPNN, we use the best-performing model as reported by the authors (Dauparas et al., 2022), trained with 0.01Å noise on coordinates. For inference, we use a sampling temperature of 0.1 and input the backbone coordinates without adding noise.

For protein docking, we retrained each method using the same splits, describe in appendix F.2. All methods were retrained using the exact parameters described in the corresponding manuscripts. Additional instructions for training and inference were gathered through correspondence with the authors of DOCKGPT and DIFFDOCK-PP. THe implementation of DOCKGPT was modified slightly to use 1Å width bins for pairwise distance features (original paper used 2Åbin-width). This was done to improve performance on rigid docking.

## H EXTENDED RESULTS

We show some additional results for protein docking in table H.1 and fig. H.1, and for Inverse folding in table H.2. For inverse folding, we experimented with low-temperature sampling as described by (Ingraham et al. (2022) Appendix, Section B).

| | I-RMSD↓ | | | L-RMSD↓ | | |
|---|---|---|---|---|---|---|
| | 25 | 50 | %≤ 3Å↑ | 25 | 50 | %≤ 6Å↑ |
| Diffdock-PP (5)[†] | 4.88 | 8.67 | 15.3% | 11.57 | 19.78 | 12.6% |
| Diffdock-PP (10)[†] | 3.87 | 6.58 | 20.0% | 9.17 | 16.49 | 15.3% |
| DIFFDOCK-PP (20)[†] | 2.63 | 5.01 | 31.3% | 6.16 | 13.20 | 24.6% |
| OMNIPROT (5)[†] | 1.65 | 2.37 | 53.3% | 3.41 | 5.53 | 50.7% |
| OMNIPROT (10)[†] | 1.54 | 2.05 | 58.0% | 2.72 | 4.36 | 54.7% |
| OMNIPROT (20)[†] | **1.45** | **1.92** | **64.7%** | **2.61** | **3.82** | **60.7%** |

Table H.1: **Results for DIFFDOCK-PP and OMNIPROT with varying number of samples** For diffusion models, the number of sampled poses is shown in parentheses. In an effort to fairly compare our method with Diffdock-PP, we report only oracle statistics, denoted with [†], which refers to the setting where we can perfectly select the best pose out of the sampled ones.

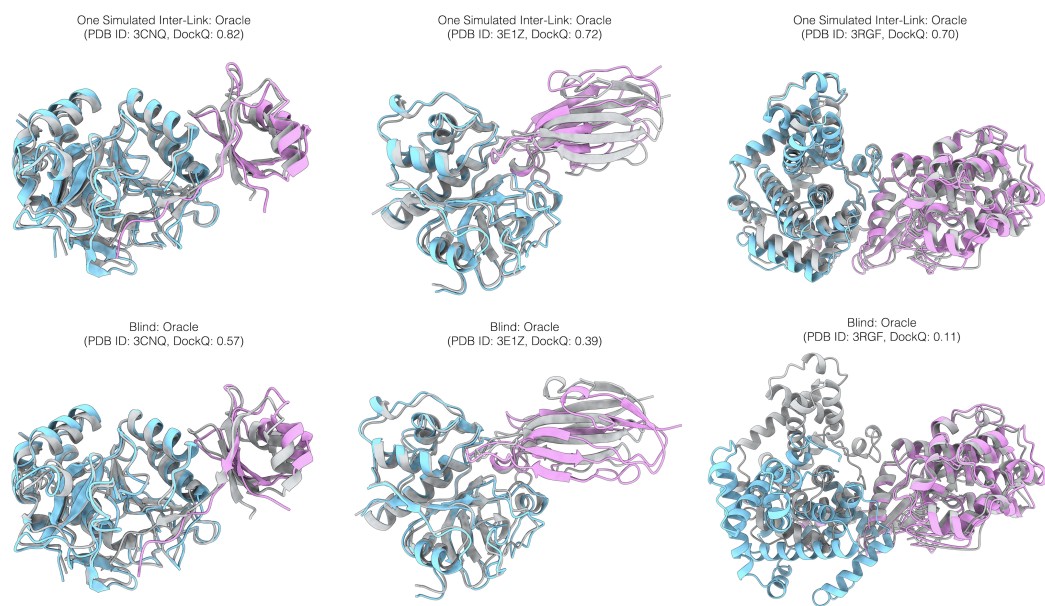

Figure H.1: **Protein-Protein Docking with OMNIPROT for three complexes** with PDB identifier: 3CNQ, 3E1Z, 3RGF from left to right. Top row shows OMNIPROT Oracle docking predictions (40 sampled poses) with contact information (one simulated inter-link). Bottom row shows OMNIPROT Oracle docking predictions (40 sampled poses) without additional contact information (blind). The respective ground truth structures are displayed in gray.

| | NSR(%)↑ | | sc-RMSD↓ | |
| --- | --- | --- | --- | --- |
| | Median | Std | Median | %≤ 2.5Å↑ |
| PROTEINMPNN (t=0.1) | 48.8% | 0.08 | 1.82 | 62.3 % |
| OMNIPROT (Seq, $\lambda = 1$) | 42.3% | 0.06 | 3.00 | 42.2% |
| OMNIPROT (Seq, $\lambda = 2$) | 43.2% | 0.06 | 2.65 | 45.8% |
| OMNIPROT (Seq, $\lambda = 4$) | 44.4% | 0.05 | 2.91 | 43.1% |
| OMNIPROT ( $\lambda = 1$) | 45.3% | 0.05 | 2.44 | 51.0 % |
| OMNIPROT ( $\lambda = 2$) | 46.8% | 0.06 | 2.34 | 53.6 % |
| OMNIPROT ( $\lambda = 4$) | 46.3% | 0.06 | 2.29 | 53.6% |

Table H.2: **Inverse Folding Results on PDB test targets, PMPNN Test Set**. Results for OMNIPROT trained to recover sequence encodings (+Seq) and joint sequence-structure encodings with varying sampling temperatures $\lambda$.

