# OpenReview forum: "Bridging Sequence and Structure: Latent Diffusion for Conditional Protein Generation"
_ICLR.cc/2024/Conference — Submitted to ICLR 2024_

### Official Review · Reviewer_b3vX · 2023-10-18

**Soundness:** 3 good
**Presentation:** 3 good
**Contribution:** 2 fair
**Rating:** 3
**Confidence:** 5

**Summary:**

In this paper, the authors present OMNIPROT, a novel generative modeling technique to capture the joint distribution between protein sequence and three-dimensional structure.

**Strengths:**

1. The method appears to be reasonable.
2. The paper is well-organized.
3. The research problem is meaningful for researchers in the computational biology field.

**Weaknesses:**

1. There are style issues, such as: `"FOr example, In drug development, ..."`, where some words have inconsistent capitalization.
2. The experiments are not comprehensive:
   - While the paper in the method section claims that the proposed method can support various protein-related tasks, this is not adequately demonstrated in the experimental section.
   - For the tasks that have been compared, the baselines are not comprehensive. For example, in the protein-protein docking task, traditional protein docking methods are not compared, and for the inverse folding task, only ProteinMPNN is compared.
3. The paper mentions that the method supports "protein-protein docking with contact information" tasks, which may not be very practical, as in real-world applications, blind docking is more common.
4. The method's innovativeness is limited. This method essentially involves multi-modal data input and the use of latent diffusion.

**Questions:**

1. The paper mentions that the encoder is trained separately, but the authors do not explain the reason for doing so. Is it because training the encoder and diffusion module together would be challenging due to the complex model architecture?
2. I hope the authors will compare their method with more baselines and also validate the model's performance on protein folding tasks.
3. For the docking task, for methods based on generative models, such as the method proposed in this paper and the compared baseline DiffDock, the paper reports "oracle" statistics by selecting the prediction with the lowest RMSD from the ground truth among the 20 generated results. I believe it would be valuable to also present the mean and variance of these 20 samples on each metric. In real-world applications, we do not have ground-truth complex structures as a reference.

---

> ### Author Response · Authors · 2023-11-23
> **Response**
>
> Your time and constructive comments are sincerely appreciated in our efforts to improve our paper.
>
> Please see the general response above regarding additional empirical evaluations and baselines.
>
> [_Docking with contact information may not be very practical._] We emphasize that our approach also readily performs blind docking, as explained in section 4 in the paper, and evaluated in the experiments section. While contact information may not always be available in practical applications, there are experimental cases where it may be (e.g. a given pocket is known). We believe that studying how the method can use this additional information and how it affects its performance to be an interesting direction, and that’s why we include it in addition to blind docking.
>
> [_Limited innovativeness._] While we respect the reviewer's perspective, we respectfully disagree and remain confident that our paper offers valuable insights and innovative contributions to the field. The unique combination of the components of OmniProt and the seamless application to diverse biological tasks like protein-protein docking and inverse folding is novel and innovative. This is aligned with a lot of work in this domain, which also applies diffusion models (over euclidean spaces or manifolds) for different protein generation tasks. We believe our work has the potential to shift the field by demonstrating the versatility of latent diffusion in addressing complex challenges in structural biology.
>
> [_Why is the autoencoder trained separately?_] We did not explore training the autoencoder jointly with the latent diffusion. Training separately is the commonly adopted choice in practice (e.g. [1]), and appears to lead to better empirical results and simpler training. While joint training of autoencoder and latent diffusion has been explored (e.g. [2]) this requires some care during training, and to our knowledge is not that commonly used in practice. Additionally, training separately reduces memory requirements (both the encoder and the score network contain triangular updates from AlphaFold, which have high memory requirements.)
>
> [1] Rombach et al. “High-Resolution Image Synthesis with Latent Diffusion Models”
>
> [2] Vahdat et al. “Score-based Generative Modeling in Latent Space”.
>
>
> [_Docking benchmark use of oracle metrics, mean and variance._] Using oracle statistics to compare generative models is common practice (see, for instance, DiffDock) as it offers insights into the upper limit of a method’s prediction accuracy, particularly when comparing to other baselines, ensuring a fair assessment of each method. For real-world generative modeling processes, it is common to introduce discriminative oracles as a substitute for known true conformations.

---

> > ### Comment · Reviewer_b3vX · 2023-11-23
> > **Reply to the authors' response**
> >
> > Thank you for the authors' responses, but some of my concerns have not been effectively addressed.
> >
> > I believe the article should be more self-contained. If you have only performed docking and inverse folding, please do not claim that your method can effectively support various tasks, such as mentioning folding tasks without conducting experiments to support it.
> >
> > Regarding the generative modeling, the article can generate diverse docking complex structures. The experimental results presented in the article are selected from multiple results, with the best-scoring complex relative to the ground-truth metrics. What I mean is that when biologists perform practical operations, there is no such thing as a ground-truth. So how can we choose a good result? Therefore, I hope the authors can show the mean and variance of multiple results to indicate whether the generated results are generally of high quality or if the quality varies significantly.
> >
> > I feel that the experiments are not detailed enough, and many baseline comparisons have not been provided by the authors. There is a lack of a baseline for inverse folding, and ProteinMPNN is not state-of-the-art. Another reviewer mentioned PiFold, which was accepted by the previous ICLR, but the authors did not provide a comparison. Furthermore, for the results on inverse folding, OMNIPROT did not demonstrate superiority.
> >
> > Regarding the docking task, existing ML-based papers have mostly reported results from traditional docking software, such as HDock. However, the authors have not supplemented this part of the experiment during the rebuttal period.
> >
> > Therefore, based on the authors' current response, I will maintain my original score for now.

---

### Official Review · Reviewer_ZFyz · 2023-10-21

**Soundness:** 2 fair
**Presentation:** 2 fair
**Contribution:** 1 poor
**Rating:** 3
**Confidence:** 5

**Summary:**

This paper constructs a unified latent diffusion framework for generating both protein sequences and structures. It leverages the proposed model for protein docking and inverse folding tasks. Experiments show that the proposed framework outperforms protein-protein docking baselines on intermediate-accuracy metrics, though not on highly-accurate metrics. It achieves similar performance to ProteinMPNN for the inverse folding task.

**Strengths:**

- The work presents a unified protein generation framework capable of performing multiple tasks.
- The paper is well-constructed, and it provides relatively comprehensive implementation details for the model and the experimental tasks.
- In Appendix F.1, the authors point out the potential data leakage issues in the commonly used DIPS dataset, which can be one of the contribution.

**Weaknesses:**

Main concerns are listed as follows:

- Although the proposed model is claimed to "bridge sequence and structure," Table 3 shows that the proposed method underperforms the inverse folding baseline on all metrics. This raises questions about the effectiveness and motivation of the unified framework by modeling both seq-structure as claimed by the authors. In the docking evaluation, OMNIPROT lags behind the baseline DockGPT by a significant margin for high-quality metrics (25th quantile for RMSD and ≥high for DockQ). This suggests that OMNIPROT struggles to recall the most accurate "hits" but generates many just "acceptable" docked poses. In traditional docking, candidate poses are searched and ranked by free energy, with only a few top (low energy) poses selected for downstream analysis or tasks. Therefore, the high-quality metrics for docking should be given the utmost consideration during evaluation, as the results weaken OMNIPROT's performance. The authors are welcome to argue against this if they disagree.
- The paper lacks an explanation for the design choices made for the OMNIPROT architecture. There is also no ablation study on the proposed framework, encompassing the encoder, decoder, and denoising network. Additionally, the authors should clarify why they adopted latent diffusion instead of a vanilla VAE or other sampling strategies used in VAE, providing corresponding evidence or support, such as an ablation study.
- While Section 5 mentions that the training data were collected "without any quality-based filtering," the authors should clarify whether they performed redundancy removal for the newly curated docking dataset. Since the docking task leverages both sequence and structure features, proper filtering for the training set (it is mentioned in the paper as "consists of the remaining data") based on sequence (and structure) similarity is required to prevent the potential data leakage.
- A related work, PROTSEED [1] (I found it already published in ICLR 2023, thus cannot be counted as concurrent work), also employs the IPA module of AF2 to decode both sequence and structures for multiple tasks, including inverse folding. The authors should clarify how OMNIPROT differs from PROTSEED and provide a better context for the reference. Failing to show significant improvement over the related works can weaken the contribution of the proposed model.
- The comparison with inverse folding baselines is insufficient, particularly with the introduction of the new metric "sc-RMSD." The authors should consider contextualizing more baselines that handled this task: PROTSEED[1], ESM-IF [2], PiFold [3], and the gradient-based AlphaDesign [4] to name a few. To establish the state-of-the-art (SOTA) performance, the authors should also refer to the PiFold, especially since it was claimed to outperform ProteinMPNN. An incomplete baseline comparison on a well-studied task may reduce the credibility of the results, and additional experiments are encouraged.
- Regarding the inverse folding evaluation, what is the rationale for using sc-RMSD instead of perplexity, where the latter is more commonly evaluated for inverse folding task? The sc-RMSD/TM are typically used for backbone (structure) generation models like FrameDiff [5]. Since the authors claim the proposed model to be "generative," the perplexity (ppl) on the test set is encouraged to be reported. The ppl can be more important in reflecting the performance of the inverse folding model compared to the flawed recovery metric.
- It is unclear to me whether the mentioned two tasks in this paper involves independent model training or if some (or all) of the model parameters are shared across tasks. If any parameters are shared, the authors could specify which ones. (please point to the related text if already mentioned)
- What is  the "geometrically-structured" latent space compared to the canonical latent space as mentioned in the abstract and introduction sections?

MISC:
- Section 1 Introduction - line 7, FOr → For.
- Equation (3), x1 → z1
- Section 3 deserves a further refinement.
- Section 4.4.1, I do not think this should be a subsection of 4.4.
- I found the paragraph in section H (appendix) incomplete.

[1] Shi, Chence, Chuanrui Wang, Jiarui Lu, Bozitao Zhong, and Jian Tang. "Protein sequence and structure co-design with equivariant translation." *arXiv preprint arXiv:2210.08761* (2022).

[2] Hsu, Chloe, Robert Verkuil, Jason Liu, Zeming Lin, Brian Hie, Tom Sercu, Adam Lerer, and Alexander Rives. "Learning inverse folding from millions of predicted structures." In *International Conference on Machine Learning*, pp. 8946-8970. PMLR, 2022.

[3] Gao, Zhangyang, Cheng Tan, Pablo Chacón, and Stan Z. Li. "PiFold: Toward effective and efficient protein inverse folding." *arXiv preprint arXiv:2209.12643* (2022).

[4] Jendrusch, Michael, Jan O. Korbel, and S. Kashif Sadiq. "AlphaDesign: A de novo protein design framework based on AlphaFold." *Biorxiv* (2021): 2021-10.

[5] Yim, Jason, Brian L. Trippe, Valentin De Bortoli, Emile Mathieu, Arnaud Doucet, Regina Barzilay, and Tommi Jaakkola. "SE (3) diffusion model with application to protein backbone generation." *arXiv preprint arXiv:2302.02277* (2023).

**Questions:**

Please address and clarify the questions and concerns in the Weaknesses section above.

---

> ### Author Response · Authors · 2023-11-23
> **Response**
>
> Thank you for dedicating your time and sharing extensive comments to improve our paper.
>
> Please see the general response above regarding additional empirical evaluations.
>
> [_Metrics, inverse folding and docking._] While we acknowledge the observed underperformance of our proposed model on some metrics, we believe it is crucial to highlight the specific design philosophy behind OmniProt, as a single model to tackle multiple tasks in the protein generation space. Unlike methods focused on a single "hit", OmniProt is designed as a generative model with the intent of providing a diverse set of poses. The emphasis here is on capturing the broad spectrum of feasible solutions that could be relevant in certain biological contexts rather than solely optimizing for the most accurate single pose. As can be seen in the results, it produces more acceptable and medium poses, which means that it is able to find good poses for cases where DockGPT fails. That being said, we believe OmniProt metrics may further improve once we train it with the updated autoencoder. Also, it is worth emphasizing that OmniProt significantly outperforms the previous state of the art diffusion for docking (DiffDock-PP).
>
> [_Ablation study for architecture._] The combination of individual components in OmniProt is unique, yet, the choices for both the autoencoder and diffusion model are fairly standard and extensive ablation studies are costly. Please see the general response for a more detailed discussion.
>
> [_Redundancy removal._] Thank you for pointing this out. By using the Foldseek score for clustering, which linearly combines both 3D-based structure and sequence substitution scores (Appendix F), our approach combines sequence and structure-based similarity metrics and restricts them specifically to interfaces, to provide a robust approach for a protein-protein-interaction benchmark dataset creation. As the test clusters are separate from the train clusters, the train dataset is inherently deleaked against the test dataset.
>
> [_Differences w.r.t. PROTSEED._] Thank you for pointing this out, we will add this to related work. While PROTSEED is also jointly encoding sequence and structure with conditioning it does so without employing diffusion. The formulation behind OmniProt uses diffusion models which were originally designed to sample distributions; in the context of protein design to sample ensembles of protein conformations. The use of diffusion leads to a different overall formulation in terms of training objective and sampling schemes. However, we do believe that PROTSEED provides a relevant baseline and we are working on adding it to our empirical evaluation.
>
> [_IF perplexity metric._] While more metrics may always yield valuable insights, we believe perplexity does not give any indication whether a protein sequence will fold into the desired structure, which is why we chose to report sc-RMSD, an in-silico proxy for designability.
>
> [_Model shared or re-trained for different tasks._] The autoencoder is trained once for each dataset, and corresponding diffusion models are trained for each autoencoder and dataset.
>
> [_Geometrically structured latent space vs. canonical._] The abstract and intro both refer to the geometrically structured latent space. We use “geometrically structured” because the representations are invariant to rigid body transformations of the input. As we explain in 4.4.1, this property means that OmniProt’s performance on conditional generative tasks is invariant to rigid body transformations of the input structures.
>
> [_Refining section 3 (related work)._] We are happy to expand or modify the related work section (section 3). We will add some missing papers raised in the reviews, any other comments and suggestions are welcome.
>
> [_Incomplete paragraph in Section H._] Thanks for bringing this up. We apologize for this oversight, we will fix the incomplete paragraph.

---

### Official Review · Reviewer_cpz6 · 2023-10-31

**Soundness:** 3 good
**Presentation:** 3 good
**Contribution:** 3 good
**Rating:** 6
**Confidence:** 4

**Summary:**

The authors introduce OmniProt, a method for conditional protein sequence and structure generation leveraging a pre-trained autoencoder and latent diffusion. By capturing the joint distribution of sequence and structure and controlling the conditioning, OmniProt can be used for protein-protein docking, folding, and inverse folding. The authors report competitive performance for both protein-protein docking and inverse folding, compared to popular baselines.

**Strengths:**

The key idea of performing conditional diffusion in the latent space of an autoencoder trained jointly on sequence and structure allows for a truly multitask generative model that encompasses many protein design capabilities.

**Weaknesses:**

There are some questions remaining about the capabilities (folding, co-generation) that are repeatedly highlighted but not evaluated in the paper. It is unclear if the autoencoder approach introduces a fundamental limitation to the method, or if there are straightforward paths to improving the decoder. The authors show one example of ablating the effects of joint training on sequence and structure, showing improvement in inverse folding metrics when training with structure-based features; their approach affords unique insight into the value of joint training and this point could be expanded upon with further experiments.

**Questions:**

1. Is there any particular reason why protein folding and co-generation have not been evaluated? The flexibility of the method is its major advantage, and even if it does not outperform purpose-built methods like AlphaFold2 (folding) and ProteinGenerator (co-generation), it would be interesting to see demonstrations of these capabilities. If the authors believe this is out of scope, it may be beneficial to narrow the scope of the claims and discussion in the paper to the two tasks that are considered in detail.
2. The authors repeatedly mention that improvements to the autoencoder might improve overall OmniProt performance. Do the authors believe that this is a scaling / expressivity problem, or architectural?
3. The authors highlight that OmniProt inverse folding samples are generated using reverse diffusion, rather than the randomized autoregressive scheme used in ProteinMPNN. This does intuitively seem like a major advantage, but no metrics are reported to support this claim. Can the authors provide some evidence beyond sequence recovery that more directly relates to sample quality?

---

> ### Author Response · Authors · 2023-11-23
> **Response**
>
> Thank you for your valuable assistance in refining this work. We are grateful for the time and effort you dedicated to providing feedback.
>
> Please see the general response above regarding additional empirical evaluations.
>
> [_Is the autoencoder a fundamental limitation? Ablation of architecture._] No, we do not believe it is. We are currently exploring changes to the autoencoder, and were able to improve its performance significantly. Please see the general answer for further details about updated performance. Since OmniProt was working at the limit of the autoencoder performance, we believe this will improve results (e.g. high-quality metrics for protein-protein docking). The main changes to the autoencoder were in the decoder, where we added pair feature updates and increased the number of layers.
>
> [_Experiments for further insights regarding joint seq+structure training._] We explored this for inverse folding. However, for docking all methods rely on the respective sequences as input, so it is not obvious to us how to decouple them. We agree studying this further is quite interesting, and appreciate any suggestions to do so.
>
> [_Metrics beyond sequence recovery._] We believe the main advantage of OmniProt comes from the flexibility in controlling the number of steps when generating samples. For diffusion models the number of steps is independent of the protein length, while autoregressive methods require one step per residue in the protein.

---

### Official Review · Reviewer_rb4f · 2023-11-01

**Soundness:** 4 excellent
**Presentation:** 3 good
**Contribution:** 4 excellent
**Rating:** 8
**Confidence:** 3

**Summary:**

This paper introduces OmniProt, a framework for generative protein design as well as a range of tasks such as folding, inverse folding, and docking.
The model consists of an autoencoder, that maps protein sequences and structures to a latent space, and a diffusion model that operates in that latent space. The autoencoder is trained using roto-translational invariant features used in DockGPT. The tasks considered of inverse folding or docking are then solved through conditioning of the diffusion model with the relevant structural or sequence information, and a decoding of the generated latent space samples.
The model shows good performance on inverse folding and state-of-the-art performance on docking compared to existing ML models.

**Strengths:**

This is an interesting study that takes a new approach to solve several related problems in protein design in a coherent way. The methods that are being compared, such as ProteinMPNN or DockGPT, are task-specific models and is quite remarkable that with a latent diffusion model, the authors are able to approach the performance of state-of-the-art models on both tasks. The OmniProt model is quite general and seems easily extendable into e.g. a de novo design framework.

**Weaknesses:**

A more in-depth benchmarking of the autoencoder, and how its performance varies as a function of structure and sequence complexity would be beneficial. In benchmarks of inverse folding, ProteinMPNN is used as reference but more recent models such as LM-Design or Knowledge-Design have shown improved performance. In the docking benchmark, it would be useful to see comparisons with traditional docking tools such as Zdock or Haddock.

**Questions:**

The authors should include a citation to 2305.04120 which has explored a relatively comparable approach of latent space diffusion for protein design.

---

> ### Author Response · Authors · 2023-11-23
> **Response**
>
> We thank you for carefully reviewing our work and providing comments to help improve the paper.
>
> [_More in depth benchmarking of the autoencoder._] As we described in the general response, we are currently exploring changes to the autoencoder, and were able to improve its performance significantly. Please refer to the general response for details.
>
> [_Cite 2305.04120, also latent diffusion for protein design._] Thank you for pointing this out, we will add this in the related work. In brief, the approach from “A Latent Diffusion Model for Protein Structure Generation” (2305.04120) differs significantly from OmniProt, as it only generates backbone coordinates (only C-alpha, while OmniProt also provides side-chain coordinates), and it does not generate sequences (while OmniProt does). Additionally, to our understanding, the method can only perform unconditional generation.

---

### Author Response · Authors · 2023-11-23
**General Response**

Thank you for the thoughtful reviews and valuable feedback on our paper. We address common comments about our empirical evaluation in this general response. We also provide more detailed individual responses for each reviewer.

[_Empirical evaluation and tasks._] We believe our evaluation is quite extensive, including protein-protein docking and inverse folding, two tasks at the core of generative modeling for proteins. While we agree that additional evaluations (tasks and baselines) and ablations would further improve the paper, we believe that our evaluation sufficiently showcases the strengths and versatility of OmniProt. By simultaneously addressing two critical tasks, which both have been the subject of many recent studies, this evaluation is already more comprehensive than most works in this domain focusing on single tasks.

[_Baselines considered._] We deliberately chose to evaluate OmniProt on the ProteinMPNN dataset to facilitate a fair comparison with what is accepted as a state-of-the-art method, thoroughly tested and evaluated, ProteinMPNN. For docking, we focused on machine learning based methods, trying to push the frontiers of what they can achieve. We emphasize we retrained three docking methods, and curated a new docking dataset, further displaying the thoroughness of our evaluation.

[_Autoencoder training and evaluation._] Following comments from multiple reviewers, we are currently exploring changes to the autoencoder, and were able to improve its performance significantly. The original autoencoder in the manuscript achieved a full atom complex RMSD of 1.3A, and ligand RMSD of 2.1A. The new variant we’re exploring reduces both these metrics to 1A. Unfortunately, we did not manage to finalize OmniProt training with the new autoencoder in the limited time for the discussion, so at this moment we are unable to provide updated tables and results. Since OmniProt was working at the limit of the autoencoder performance, we believe this will improve results (e.g. high-quality metrics for protein-protein docking).

Thank you once again for the thorough review, and we look forward to presenting our work with these considerations in mind.

---

### Meta-Review · Area_Chair_6SgR · 2023-12-06

**Metareview:**

This was a fairly divisive paper for the reviewers, with some reviewers feeling that the authors' method was quite interesting due to the ability of a single model to perform moderately well on two of the most popular protein design tasks (inverse folding, docking). Other reviewers felt that, at the end of the day, the model wasn't actually better at any one of these tasks despite its versatility.

I think that ultimately I agree that the authors could do a better job of highlighting the payoffs of their method, or specifically situations where one might use their model over a more specialized model. I'll give two concrete suggestions here. First, include results on additional tasks that further highlight the versatility. Reviewer b3vX's point that the authors claim broad applicability but ultimately evaluate two tasks is not wholly unreasonable. Second, if the authors find even a single reasonable task (perhaps one that is not as well studied as docking or inverse folding) for which their method out of the box produces strong performance, that would seem to me to immediately recover the method's versatility as a strength despite its underperformance on the two specific tasks considered.

**Justification For Why Not Higher Score:**

The authors could do a better job of highlighting the payoffs of their method, or specifically situations where one might use their model over a more specialized model.

**Justification For Why Not Lower Score:**

N/A

---

### Decision · Program_Chairs · 2024-01-16

Reject